# Analysis and Historical Evolution of Paediatric Bone Tumours: The Importance of Early Diagnosis in the Detection of Childhood Skeletal Malignancies

**DOI:** 10.3390/cancers17030451

**Published:** 2025-01-28

**Authors:** Giulia Iacobellis, Alessia Leggio, Cecilia Salzillo, Stefano Lucà, Ricardo Ortega-Ruiz, Andrea Marzullo

**Affiliations:** 1Department of Clinical and Experimental Medicine, Radiology Unit, University of Foggia, 71122 Foggia, Italy; giulia.iacobellis@unifg.it; 2Department of Interdisciplinary Medicine, Legal Medicine Unit, University of Bari “Aldo Moro”, 70121 Bari, Italy; 3Department of Precision and Regenerative Medicine and Ionian Area, Pathology Unit, University of Bari “Aldo Moro”, 70121 Bari, Italy; cecilia.salzillo@unicampania.it; 4Department of Experimental Medicine, PhD Course in Public Health, University of Campania “Luigi Vanvitelli”, 80138 Naples, Italy; stefano.luca@unicampania.it; 5Department of Mental and Physical Health and Preventive Medicine, Pathology Unit, University of Campania “Luigi Vanvitelli”, 80138 Naples, Italy; 6Faculty of Criminology, Isabel I de Castilla International University, 09003 Burgos, Spain; ricardo.ortega@ui1.es

**Keywords:** paediatric bone tumours, osteosarcoma, Ewing’s sarcoma, early diagnosis, diagnostic imaging

## Abstract

Malignant tumours of bone tissue in children are rare neoplasms, and identifying this pathology at an early stage is crucial for implementing an appropriate plan for effective functional therapy. Osteosarcoma and Ewing’s sarcoma are the primary paediatric bone tumours discussed in the work presented. Our review aims to provide a clearer understanding of the main diagnostic techniques, which are fundamental for identifying these malignant pathologies and enabling timely intervention, thereby ensuring the patient receives suitable treatment. The examination of bone tumours requires extensive medical knowledge and detailed diagnostics. In addition to the macroscopic description, radiological investigations such as X-rays, magnetic resonance imaging (MRI), and computerised tomography (CT) are essential to ensure an accurate diagnosis. We hope this review will assist both pathologists and clinicians in enhancing the rapid detection of paediatric malignancies through diagnostic techniques.

## 1. Introduction

Although primary bone sarcomas are a rare group of oncological diseases with an incidence of less than 1 per 100,000 per year, they are even rarer in the paediatric population. Nevertheless, studying these conditions is crucial to improving diagnostic and therapeutic processes, thereby enhancing patient outcomes [1,2].

The most frequent malignant bone neoplasms in paediatrics are the incidence of osteosarcoma, which is 4.4 incidences per million inhabitants per year, and Ewing’s sarcoma, where the annual incidence is 2.2 incidents per million in the paediatric population [2,3]. Cases of osteosarcoma and Ewing’s sarcoma have also been identified in skeletal remains from newborns (in rare instances) and young adults in various archaeological contexts. The palaeopathological analysis of these conditions is invaluable for understanding their origins, morphological changes, and evolution over time [4,5].

One type of high-grade osteogenic tumour is osteosarcoma, which is histologically characterised by an osteoid matrix and, in varying quantities, a chondroid matrix and fibrous tissue. Defects in the RB and p53 genes are important in the development of osteosarcoma; however, the precise aetiology is yet understood. The risk of osteosarcoma is almost 1000 times higher for patients with germline mutations in RB, and the same is true for patients with Li–Fraumeni syndrome (germline p53 mutation), who also exhibit a markedly higher incidence of this tumour [6]. Diagnosis involves an integration of histopathological and radiological evaluations.

The illness is more common in the second decade of life, which corresponds with pubertal development and typically arises at the metaphyseal regions of long bones, with a prevalence in males [2]. In some cases, osteosarcoma may also appear in older populations, particularly in association with Paget’s disease, where it has a peak incidence in individuals over 60 years of age.

From a palaeopathological perspective, osteosarcoma in historical human remains can be identified macroscopically by a highly fissured bone cortex and specific patterns of bone growth. The periosteum appears thickened, ossified, and often exhibits evident radiating bony growths indicative of reactive periosteal deposition associated with bone neoplasms. Radiological findings reveal early-stage bone remodelling characterised by calcifications extending perpendicularly from the bone cortex [1].

Ewing’s sarcoma is a high-grade malignancy of small, round cells, which is typified by recurrent chromosomal translocations, most commonly t(11;22), which result in the EWS-FLI1 fusion transcript. This gene product regulates target genes involved in tumour development. The disease’s epidemiological peak occurs around the age of 15, though it can present at any age, with poorer prognoses in later life. Ewing’s sarcoma affects both the skeleton and soft tissues, with a higher prevalence in males, and makes up around 6% of all primary bone tumours [6,7].

## 2. Materials and Methods

Our study is a comprehensive review of the main paediatric bone neoplasms, addressing the subject of diagnosis and treatment, emphasising the significance of markers for the early diagnosis of these tumours and the radiological diagnosis that allows both an early diagnosis and a better therapeutic methodology. Using the search terms “bone imaging in children”, “osteosarcoma”, or “Ewing’s sarcoma”, along with inclusion criteria for primary, secondary, and English-language research, the review was carried out on PubMed and Scopus. For the grey literature, Google Scholar and a few textbooks were also used.

## 3. Early Diagnosis of Osteosarcoma and Ewing’s Sarcoma

Early symptoms are important in identifying a bone tumour since they can help differentiate it from other orthopaedic disorders, including tendinitis and osteomyelitis, as well as from other bone neoplasms. The local size of the tumour and the existence of metastases are the primary factors influencing the prognosis of osteosarcoma and Ewing’s sarcoma. When metastases are present, survival chances drop by two to three times [8,9]. According to data from the US and Europe, a diagnosis is typically made one to three months after the patient first exhibits symptoms.

Reducing the time between the onset of symptoms and the diagnosis of Ewing’s sarcoma and osteosarcoma is crucial. Physician awareness of the typical early symptoms, including pain, fever, local volume increase, and, in some cases, the initial pathological fracture, is essential for this. For the purposes of differential diagnosis, it is important to consider the localisation of pain. Considering the most frequent sites of neoplastic localisation, Ewing’s sarcoma has a more uniform distribution in the femur and tibia portions and more often affects the distal portion of the fibula, in addition to more frequently affecting the pelvis and other parts of the body (like the spine). In contrast, osteosarcoma tends to affect the distal portion of the femur, the proximal portion of the tibia, fibula, and humerus, and, less frequently, the pelvis or other parts of the body. The mean age of patients with osteosarcoma is 15.7 years, whereas those with Ewing’s sarcoma are younger (mean age: 12.8). The most common early symptom (89.5% of the time) in patients with osteosarcoma or Ewing’s sarcoma is local pain. In addition to pain, these tumours also cause weight loss, pathological fracture, and local pain during rest in similar amounts. Patients with Ewing’s sarcoma are more likely to have a fever, whereas those with osteosarcoma are more likely to exhibit a volume rise. The average duration between the beginning of Ewing’s sarcoma and diagnosis is 32.4 weeks (8.1 months), while osteosarcoma is diagnosed at 21.12 weeks (5.25 months) [10].

A significant proportion of paediatric bone tumour cases have a constitutional genetic foundation (Table 1); according to some series, germline TP53 mutations are present in about 10% of OS children.

OS risk is elevated in individuals who have retinoblastoma predisposition RB1; the RB1 protein is a negative cell cycle regulator. Rarely, persons with multiple hereditary exostoses (EXT1, MIM 608177; EXT2, MIM 608210)—both genes encode glycosyltransferases involved in heparan sulfate production—may also develop chondrosarcomas later in life. Furthermore, OS was observed in patients with Diamond–Blackfan anaemia (RPS19, MIM 603474; RPS17, MIM 180472; RPS24, MIM 602412; RPL35A, MIM 180468; RPS7, MIM 603658; RPL5, MIM 603634; RPL11), RECQL4 (encoding a DNA helicase; MIM 268400), Werner syndrome, Bloom syndrome, RAPADILINO syndrome, RECQL4c (MIM 266280, Rothmund–Thomson syndrome, and RECQL4 (encoding a DNA helicase; MIM 268400) [11].

More recent studies using next-generation sequencing (NGS; whole exome sequencing, WES; and whole genome sequencing, WGS) reveal a genomic landscape predominantly characterised by somatic copy number alterations rather than point mutations/indels. Understanding the genomic and immune landscape of OS has presented an opportunity for the introduction of new molecularly targeted therapies and novel immuno-oncology approaches [12].

Biomarkers linked to OS and Ewing’s sarcoma have been found in numerous fundamental, translational, and clinical investigations. These findings suggest that a variety of biomarkers may be employed to track the course of an illness or forecast its prognosis. However, particularly in pediatric patients, there are not many signs that cancer caretakers can trust to make a confirmed diagnosis early in the course of the disease. Among the main biomarkers, we can consider the following.

MicroRNAs play a significant role in oncogenesis and have an inhibitory effect. Since they control cell proliferation, invasion, metastasis, apoptosis, and angiogenesis, a number of dysregulated microRNAs (miRNAs) have been found recently in sarcomas and are thought to be promising biomarkers for diagnosis and prognosis. Some studies [13] revealed that the levels of miR-195, as measured by real-time quantitative reverse transcription polymerase chain reaction (RT-PCR), were lower in the serum of OS patients than in normal controls, indicating a possible role of miRNA as a serum marker for OS diagnosis. Examples of miRNAs that are upregulated in OS include miR-195, miR-99, miR-181, and miR-148a, while others, like miR-539, miR-145, and miR-335, have been found to be downregulated in human OS cell lines MG-63. Regarding ES, other miRNAs that were connected to progression, survival, and therapeutic response were also found in patient samples and cell lines. A recent study proposed miR34a as a viable biomarker in the follow-up of ES disease, even though miR-185 was shown to be involved in ES cell proliferation, motility, and survival by downregulating the PI3K/Akt/mTOR and Wnt/β-catenin pathways, which functions as a tumour suppressor. Additionally, it was demonstrated that the EWS-FLI1 fusion protein, which is commonly seen in ES, changes miRNA expression [14].

The immune markers proinflammatory cytokines, including interleukin-6, protein-1, tumour growth factor-β, growth-related oncogene, chemokine ligand 16, endoglin, matrix metalloproteinase-9, and platelet-derived growth factor-AA, are abnormally elevated in plasma levels linked to OS. Shang and colleagues discovered that TIM3 was expressed in nine invasive human OSs [13]. OS patients had higher serum levels of galectin 3 than healthy controls, and advanced stages showed progressively higher levels of the protein. Abnormal TIM3 expression has been linked to carcinomas.

Other protein markers, including hypoxia-inducible transcription factor (HIF-1), HIF, VEGF (vascular endothelial growth factor), and purinic/apyrimidinic endonuclease 1 (APE1), are involved in the DNA base excision repair pathway and have also been discovered to be increased in the OS cell line in response to hypoxia. However, these proteins are over-expressed in many neoplastic diseases and are, therefore, not specific in the diagnosis of these bone tumours [15].

## 4. Clinical Features

Persistent pain at the site of the neoplasm often prompts the patient to undergo an X-ray investigation. The condition is associated with a poor prognosis due to rapid metastasis to the lungs, necessitating a chest X-ray or chest CT for further evaluation.

The role of serological markers, such as alkaline phosphatase (ALP) and lactate dehydrogenase (LDH) is fundamental in the follow-up of lesions. Elevated levels of ALP are commonly observed in cases of osteosarcoma due to increased osteoblastic activity [6].

A definitive diagnosis is achieved through biopsy, which confirms the diagnosis and determines the tumour grade. Clinically, Ewing’s sarcoma is distinguished by the presence of fever, leukocytosis, elevated erythrocyte sedimentation rate (ESR), and anaemia. These symptoms, in conjunction with common manifestations such as pain and swelling, represent an unfavourable prognostic sign and may be mistakenly attributed to osteomyelitis [16].

Both diseases can be differentiated by their distinctive characteristics, as summarised in Table 2.

## 5. Imaging Techniques in the Evaluation of Bone Neoplasms

Radiography, CT, MRI, bone scintigraphy, and PET CT/PET MRI are among the multimodality approaches frequently employed in the diagnostic work-up of bone malignancies [18].

Each modality provides a unique contribution to the evaluation of bone tumours. When combined in various configurations tailored to the specific disease process, they offer an accurate roadmap for radiographic diagnosis and treatment of bone tumours (Table 3).

To investigate bone discomfort or swelling, radiographs are typically obtained in two planes: anteroposterior (AP) and lateral/oblique. Conventional radiography remains the most effective method for evaluating primary bone cancers. Its cost-effectiveness and the distinct advantage of 2D imaging allow for the characterisation of lesions based on radiographic features.

Plain radiography can reveal the location of the lesion within the bone and provide key tumour imaging characteristics, including margins, edges, matrix mineralisation, cortical involvement, and periosteal reactions [16,20,21]. Multidetector CT offers detailed anatomical delineation and is particularly useful for evaluating lesions in complex anatomical regions where radiographs may be insufficient due to low-contrast resolution. CT scans are highly effective in detecting subtle bone abnormalities, calcifications, tumour mineralisation, cortical changes, and periosteal responses.

MRI is the preferred method for local staging of bone cancers. It is the primary imaging modality for evaluating suspected or diagnosed bone tumours due to its ability to visualise soft tissues and bone marrow. MRI facilitates the accurate identification of bone cancers by assessing tissue composition, including the involvement of soft tissues and bone marrow. Advanced MRI methods, including magnetic resonance spectroscopy, diffusion-weighted imaging (DWI), and dynamic contrast-enhanced imaging (perfusion MRI), further enhance its diagnostic capabilities.

When it comes to tissue characterisation and bone tumour staging, functional magnetic resonance imaging (fMRI) offers a number of benefits over structural MRI [22]. While standard MRI evaluates structural alterations and the extent of disease, diffusion-weighted imaging (DWI) and perfusion MRI allow for the analysis of tumour morphology and the differentiation between benign and malignant causes. Standard MRI alone cannot determine tissue cellularity, viability, or necrosis—key criteria for post-chemotherapy follow-up assessment—making perfusion MRI and DWI essential.

Perfusion MRI involves plotting the enhancement pattern against a time curve, with early enhancement used to evaluate tissue vitality and vascularisation inside the tumour. Dynamic MRI can monitor changes following chemotherapy by distinguishing viable tissue from fibrous tissue, scarring, or necrosis, and it can guide biopsy by detecting viable tumour regions.

DWI enhances structural evaluations by qualitatively assessing tissue cellularity. Malignant tumours, which typically exhibit higher cellular density, show limited dissemination. Diffusion-weighted pictures show signal reversal and limited diffusion on the apparent diffusion coefficient (ADC) in densely cellular tissues, reflecting reduced Brownian movement—a key principle of DWI.

Treatment response is evaluated through a multimodality approach, incorporating X-rays and MRI. Standard MRI techniques offer a comprehensive view of tumour response, and when no visible change in lesion size is observed, assessing tissue viability becomes critical for determining treatment efficacy. Functional MRI complements morphological imaging by providing additional, interpretable data.

Dynamic contrast imaging of bone tumours reveals early and rapid progressive enhancement, indicative of residual or recurrent tumours. Conversely, the absence of early enhancement suggests an excellent treatment response. DWI, with ADC calculation, further distinguishes viable from non-viable tissue components by identifying restricted diffusion in residual tumours [23,24].

## 6. Osteosarcomas

### 6.1. Pathological Subtypes

The World Health Organization has identified eight subtypes of osteosarcoma of bone, each with unique biological characteristics and clinical outcomes: small cell, low-grade central, secondary, paraosteal, peri-osteal, high-grade surface, conventional, and telangiectatic [6].

Conventional osteosarcoma represents the majority of cases, approximately 80%, and is characterised by osteoid production by malignant mesenchymal cells. Within this category, three main histological subtypes are identified: osteoblastic, dominated by bone matrix formation; chondroblastic, characterised by the significant presence of tumoural cartilage; and fibroblastic, which shows a prevalence of fibrous tissue and a reduced quantity of osteoid. These histologies are all characterised by a high degree of cellular pleomorphism and mitotic activity.

Telangiectatic osteosarcoma represents approximately 4% and is characterised by the presence of large cavities filled with blood, separated by septa containing malignant tumour cells and osteoids. Differential diagnosis from cystic bone aneurysm is essential, as both lesions may appear similar on radiological imaging.

Small cell osteosarcoma accounts for approximately 1–2% and is characterised by small tumour cells with hypochromatic nuclei, and the presence of osteoids allows the differential diagnosis from Ewing’s sarcoma.

Low-grade osteosarcoma is another less frequent, generally more insidious histotype, which presents slow growth and less cellular atypia compared to more aggressive forms but can evolve into high-grade tumours if not adequately treated. Histologically, it may be mistaken for benign lesions such as fibrous dysplasia or desmoplastic fibroma.

Finally, there are superficial forms of osteosarcoma, such as the parosteal, low grade, which originates from the bone surface with well-organised bone trabeculae; the periosteal, of intermediate grade, with a mainly cartilaginous matrix and calcifications; and high-grade superficial osteosarcoma, an extremely rare variant that shares characteristics with conventional osteosarcoma but located on the bone surface.

The histotypic variety highlights the importance of an accurate histopathological diagnosis, combined with radiological and molecular techniques, to identify the subtype and plan the best treatment.

The most common sites of osteosarcoma involvement are the metaphyses of long bones (91%), with the diaphysis being less frequently affected (9%). It has a marked predilection for the distal metaphyseal or meta-diaphyseal regions of the femur and the proximal regions of the tibia and humerus. In later life, osteosarcoma most commonly involves the knee, mandible, pelvis, scapula, spine, and skull. Involvement of the elbow and ankle is exceedingly rare [13,25,26].

### 6.2. Imaging

The first-level investigation is represented by the X-ray, in which the radiographic aspect is that of a malignant tumour that originates inside the bone, rapidly destroying the cortical, the nascent sol appearance, the periosteal lift with the formation of the Codman’s triangle, the apposition of new bone in the tissues (Figure 1a,b).

After chemotherapy, the tumour becomes well-defined, encapsulated, and more mineralised. Depending on the osteogenic entity, it can manifest itself in pure osteolytic form, pure thickening form, and mixed form (more frequent). Radiologically, in the mixed form, the association of dense osteosclerotic lesions with defined margins, continuous or nubecular-like, and confluent osteolytic lesions with “mouse bite”, is observed.

The medullary canal is often involved (Figure 2). The growth of cartilage or physis can hinder its spread to the epiphysis.

Sarcomatous osteogenesis induces the appearance of linear, coarse, thick “sunray” calcific beams implanted perpendicular to the axis of the bone (Figure 3).

In the pure thickening or osteolytic forms, the semiological criteria are the same, with poor periosteal reactions in the former and intense reactions in the latter.

CT allows the evaluation of the cortical integrity (small osteolytic areas of compact) and alterations of the cancellous trabecular meshwork, but above all, it demonstrates fine tumour calcifications (Figure 2).

MRI is very accurate in assessing the degree of extension of the tumour of the medullary canal and paraosteal soft parts from the presence of small bone metastases at a distance from the primary lesion without, however, allowing a reliable demonstration of tumour calcifications.

The most sensitive imaging technique for identifying and defining bone tumours, particularly when the medullary cavity is involved, is magnetic resonance imaging (MRI). Since MRI is the most effective local staging modality, it can be useful in the preoperative evaluation of patients whose conventional radiographs show an aggressive bone lesion (Figure 4).

Even in the face of a negative bone scan, MRI can allow the discovery of occult intramedullary lesions.

MRI is essential for the following:
The evaluation of intraosseous tumour extension with T1-weighted or STIR sequences or in the evaluation of “skip lesions” (for example, a smaller lesion inside the same bone that is physically distinct from the main tumour due to normal marrow intervening);The evaluation of extraosseous tumour extension (assessed with T2-weighted or STIR images); in these sequences, the tumour becomes hyperintense compared to fat;Evaluation of neurovascular or joint involvement, which is essential in understanding, if possible, a conservative surgical or amputation treatment;Evaluation of local and/or regional lymph nodes because lymph node involvement leads to a poor prognosis as well as distant metastases;Tumour evaluation after therapy is fundamental in the prognostic evaluation of the patient. If viable tumour cells make up less than 10% following treatment, this suggests a positive reaction, while greater than 10% at the post-treatment lesion indicates a poor response. The response to chemotherapy involves a reduction in neo-angiogenesis and necrosis and a volumetric reduction in the neoplasm with capsulation. Both tumour and reactive non-neoplastic tissue show enhancement on standard T1-weighted post-gadolinium examination; this technique is not used for this purpose.

The aspects of dynamic enhancement with gadolinium, obtained with rapid T1-weighted post-gadolinium sequences, showed a high degree of correlation with responses or non-responses as the residual tumour presents enhancement earlier than reactive tissue. Surgery or radiation therapy often results in intra-tissue oedema or haemorrhage, but the absence of a defined mass is firm evidence that there is no tumour recurrence.

This is evaluated in T1-weighted pictures by examining the intermuscular fascial planes for distortion or the disappearance of the typical marbling of muscle fat; the usual musculoskeletal architecture in the T1-weighted images is highly predictive of the absence of tumour recurrence, even though the signal intensity is present in the T2-weighted pictures or improvement following gadolinium injection. A biopsy is recommended if a mass with post-contrast impregnation is found since it is most likely a relapsing tumour; however, post-therapy granulation tissue may take on the same characteristics [27,28,29].

PET/CT with 18F FDG, in addition to delineating the extent of the primary tumour, highlights the presence of extra-skeletal metastases with great sensitivity. The entire body is often imaged using osseous scintigraphy with technetium 99m (99mTc) methylene diphosphonate and/or PET using fluorine 18 (18 F) fluorodeoxyglucose. Nonetheless, there has been a change in favour of PET systems.

The availability of PET/MRI hybrid systems in recent years has led to the additional benefits of lowering the patient’s overall radiation dose and representing crucial soft tissue details, even though PET has historically been used in conjunction with TC to provide anatomical correlation [30,31].

### 6.3. Management

For successful treatment of osteosarcoma, a multidisciplinary team comprising paediatric oncologists, radiologists with musculoskeletal expertise, oncological orthopaedic surgeons, and pathologists are required. Historically, ablative surgery with wide amputation was the most common treatment; however, long-term disease-free survival was rare. The introduction of multimodal chemotherapy in the 1970s significantly improved prognoses. Advanced limb-sparing procedures have replaced primary amputations, leading to enhanced quality of life and function following surgery while maintaining local disease control rates [32,33].

Patients with conventional osteosarcoma often undergo two courses of neoadjuvant chemotherapy before surgery [34]. A positive response to therapy is linked to a greater proportion of chemonecrosis, as observed in histopathological examinations. A good response to treatment is frequent, along with less tumour development and better clinical signs. Unlike soft-tissue tumours, osteosarcomas rarely exhibit significant shrinkage. Instead, numerous clinical studies have shown that decreased growth rates and intralesional ossification function as surrogate indicators of therapeutic response (Figure 5).

In serial radiographs, progressive mineralisation signifies a favourable reaction, while continuous bone loss denotes the advancement of the disease. Chemotherapy is necessary for treating suspected micrometastases, but it is not enough for managing local tumours; “en bloc” excision of the primary tumour is required. Pulmonary wedge resection is also tried in patients with lung metastases. After the surgical incision has healed, usually a few weeks after definitive surgery, post-operative (adjuvant) chemotherapy is started. The existence of distant disease, the feasibility of surgically controlling the tumour locally, and the response to neoadjuvant treatment all influence the exact time and drug schedule for adjuvant chemotherapy [36,37,38,39].

Surgical treatment for osteosarcoma involves tumour excision and limb reconstruction [40]. The Enneking classification categorises surgical tumour margins into intralesional, marginal, wide, and radical. Because local recurrence rates are higher, intralesional and marginal margins are typically avoided. In up to 95% of cases, wide surgical margins allow for local disease control and are therefore favoured over radical margins since they are less incapacitating. Depending on anatomical location, growth pattern, and closeness to critical tissues, different criteria are used to define a wide surgical margin. Amputation is often performed in cases of neurovascular involvement and when limb-sparing procedures would result in limited functionality. Surgical reconstruction is complex, and the child’s age and remaining growth potential must be considered.

Currently, there are few novel therapy options available for osteosarcoma, particularly for recurrent and metastatic cases. However, advances in biomarker development and a better understanding of the immune response to osteosarcoma have led to more patients benefiting from immunotherapies in recent years. There are still issues to be resolved, such as determining which immunotherapies and checkpoint inhibitors work best, lowering the toxicity of cancer vaccines and cytokines, and preventing paradoxical or hyperprogressive disease [33].

Chimeric antigen receptor (CAR)-T cell therapy, an emerging immunotherapy approach that targets tumour antigens and releases immune factors, has made substantial progress in treating haematological cancers and, to a limited extent, solid tumours, including osteosarcoma. Its efficacy is limited by factors such as low antigen specificity, short persistence, and a complex tumour microenvironment [36,37]. Chemotherapy and surgical resection are two contemporary treatment options for osteosarcoma that have increased survival rates for non-metastatic cases but are still insufficient for recurrent or metastatic cases. Oncolytic viral therapy (OVT), which uses naturally occurring or genetically modified viruses to precisely lyse cancer cells and trigger a strong immune response against remaining osteosarcoma cells, is a promising substitute [41,42].

To sum up, standard radiographs, an MRI of the main tumour, a chest CT scan, and whole-body imaging should all be part of the staging process. To prevent misclassifying the response to neoadjuvant therapy, imaging should be carefully performed within 28 days of starting chemotherapy [19,43,44,45].

## 7. Ewing Sarcoma

### 7.1. Pathological Subtypes

Mesenchymal neoplasms, including small round-cell sarcomas, present diagnostic challenges due to their rarity, overlapping features, and difficulties in interpreting molecular tests. The recent WHO classification identifies four main categories: Ewing’s sarcoma, round cell sarcoma with non-ETS EWSR1 fusions, sarcoma with CIC rearrangements, and sarcoma with BCOR alterations [27,46,47].

Ewing’s sarcoma is a small, round-cell malignant tumour with well-defined macroscopic and histological characteristics. Macroscopically, it is usually a solid, yellowish-white mass, which may include central necrosis and areas of haemorrhage, often with a torn tissue appearance due to rapid growth.

Histologically, Ewing’s sarcoma is characterised by monomorphic round cells with scant cytoplasm and uniform nuclei, arranged in sheets with minimal stroma, but larger cells with irregular nuclei and, rarely, a large cell variant with marked atypia called atypical Ewing may be observed. Some cases show rosette-like formations and a “light cell–dark cell” pattern. Mitosis and necrosis are variable. Atypical features such as alveolar architecture, myxoid or hyalinised stroma, and spindle cell morphology, while rare, suggest alternative diagnoses.

In immunohistochemistry, Ewing sarcoma has strong, diffuse membrane positivity for CD99 and, in 90% of cases, is positive for NKX2.2.

Genetically, Ewing’s sarcoma is characterised by fusion of the EWSR1 genes, typically with the ETS gene, such as FLI1 in 90% or ERG in 5%; mergers with rarer partners are FEV, ETV1, and ETV4, but detection of these mergers alone is not diagnostic [18,48,49].

These combined approaches of macroscopic, histological, immunohistochemical, and molecular study are essential for a correct diagnosis of Ewing’s sarcoma to avoid confusion with other small, round-cell sarcomas.

Ewing’s sarcomas typically develop in soft tissues rather than bone, and approximately 25% of patients have identifiable metastases upon diagnosis. Metastases most commonly occur in the lungs (50%), followed by bone (25%) and bone marrow (20%) [50,51,52].

It can occur in any bone but most frequently affects the long bones in 60% with diaphyseal and meta-diaphyseal sites, pelvis, shoulder girdle, and cyst, while it is rare in the tubular bones of the foot and is exceptional in the hands.

Eighty-six percent of cases include the ribs, extremities, and pelvis. The spine is affected in 4–6% of cases, and other rare locations include the scapula, 4–5%; hand/foot bones, 4–6%; radius/ulna, 3–5%; mandible/maxilla, 1–2%; clavicle, 1%; facial bone, 0.5%; and sternum 0.2%. Diaphyseal and metaphyseal lesions predominate. The rare location of origin is epiphysis 0.5–2%, while 10% of cases have an extension to epiphysis. Previous trauma often recurs in the anamnesis. Clinically, it can simulate osteomyelitis [53,54].

### 7.2. Imaging

The first-level diagnosis is represented by X-ray; however, to identify the loco-regional and distant extension more accurately, CT and MRI evaluation is necessary. Ewing sarcoma is very radiosensitive.

Spiral CT, whole-body MRI, and FDG-PET are frequently used, although further research is needed to determine how they affect patient classification and prognosis.

The current gold standard for evaluating primary tumours is MRI. Although whole-body MRI is sensitive in identifying metastases, its use is constrained by time constraints, particularly in children. According to a pilot research, quick whole-body MRI with echoplanar and turbo short T1 inversion–recovery sequences was just as successful as more conventional techniques, such as bone scans, CT, MRI, and ultrasonography, in identifying metastases in children with sarcomas. The radiographic appearance is variable and not always characteristic: the lesional pattern is essentially osteolytic as the tumour grows, invading the canals of Havers and the medullary spaces. The cortical is moth-eaten with an important periosteal reaction; in 50%, it is a multilamellar “onion bulb”, while it is rarely spicular with the formation of the Codman triangle [55,56].

A broad zone of transition and a lamellated/speculated periosteal reaction are characteristics of lesions that are usually aggressive, destructive, or moth-eaten (Figure 6a,b).

Radiologically, the most common form is characterised by multiple osteolytic areas, ranging from small to non-confluent foci. If the compact bone is affected, it may appear “flaky” due to progressive lamellar dissociation and, in advanced stages, it can disappear.

Frequent (58–84%) and generally violent (94%) periosteal response frequently appears as either spiculated or lamellated (“onion-skin”), (“sunburst” or “hair-on-end”).

The pathological differential diagnosis includes osteomyelitis, eosinophilic granuloma, osteosarcoma, non-Hodgkin lymphoma (NHL), and bone metastases from neuroblastoma. More rarely, it presents as an isolated, poorly delimited osteolytic form. Soft tissue invasion is a constant feature.

Although periosteal Ewing sarcoma without medullary canal involvement is uncommon (3% of cases), it can lead to extrinsic bone degradation and may resemble periosteal osteosarcoma on radiographs.

Similar to its radiographic presentation, Ewing sarcoma appears on CT as severe bone degradation with a sizable soft-tissue tumour. The pattern of intracortical involvement is generally undetectable on CT and requires examination using large window settings. For 30% of individuals, intracortical involvement is the only sign of cortical injury. In 17% of cases, there is no discernible link between the soft-tissue components and the medullary canal, and the cortex shows up on CT as intact. Contrast enhancement is prevalent on CT, and in our experience, it is typically diffuse or peripheral nodular.

Due to its superior contrast resolution, MRI is the most appropriate radiological technique for evaluating bone and soft-tissue tumours, including Ewing sarcoma.

MRI of bone Ewing sarcoma shows 100% marrow replacement and 92% cortical damage, with a soft-tissue tumour evident in 96% of patients. The associated soft-tissue mass is often circular but asymmetric with respect to the osseous involvement (Figure 7).

Signal intensity on T1-weighted images is often moderate (95%) or homogeneous (73%). Ewing sarcoma frequently appears homogenous (86%), with low to intermediate signal intensity (68%) on T2-weighted imaging.

The high level of cellularity in Ewing sarcoma is most likely the cause of this signal’s strength and uniformity. On T1-weighted images, heterogeneity is seen in 27% of cases, whereas on T2-weighted images, it is seen in 14% of cases. On T2-weighted images, 32% of Ewing sarcomas of bone display high signal intensity (Figure 7c).

In our experience, bigger lesions are more likely to exhibit heterogeneity and high signal intensity on long TR pictures, which suggests haemorrhage or necrosis. Because of haemorrhage, fluid levels can also be seen; however, this is more commonly seen following therapy.

In CT or MR imaging, the pattern of cortical involvement and destruction suggests a tiny, blue-cell tumour, such as leukaemia, lymphoma, or Ewing sarcoma. Four percent of the time, MR imaging shows an intact cortex between the medullary canal and soft tissue. Subperiosteal soft-tissue mass is characteristic of Ewing sarcoma within the periosteum, which does not involve CT or MR imaging of the medullary canal.

MR imaging of Ewing sarcoma consistently shows contrast enhancement, which can be diffuse or peripheral nodular in nature [57,58,59].

### 7.3. Management

Ewing’s sarcoma patients usually exhibit symptoms associated with the tumour, such as pain or a palpable lump.

Imaging the suspected cancer is the initial stage in the evaluation procedure, preferably using MRI to encompass the entire affected bone or compartment. This should ideally be performed before any bleeding or swelling caused by the biopsy. The biopsy should be performed or assisted by the surgeon who performs local control surgery whenever feasible.

Maintaining future reconstructive choices depends on the biopsy tract being positioned precisely and tissue planes being preserved. Open biopsies are gradually being replaced with closed, imaging-guided core biopsies. However, close collaboration between surgeons and pathologists remains essential to ensure diagnostic success.

Once the diagnosis is confirmed via biopsy, neoadjuvant chemotherapy is typically initiated as the first-line treatment to eradicate micrometastases and reduce the primary tumour, as demonstrated by several clinical studies (Figure 8).

Ewing’s sarcomas often exhibit significant necrosis, necessitating adequate tissue sampling for immunohistochemical analysis. Because Ewing’s sarcomas are radiation-sensitive, over the past 30 years, fewer individuals have received irradiation as their primary treatment. This trend is attributed to advancements in orthopaedic surgery and a growing awareness of the long-term adverse effects of radiation on children, including secondary growth abnormalities and cancers.

Higher survival rates are linked to the main tumour being surgically removed; however, prognostic factors such as tumour location and size complicate survival analysis.

Conventional cytotoxic treatments are ineffective in approximately 25% of patients with localised tumours and in 75% of patients with metastases [56,57].

New therapeutic approaches have been explored, starting with the discovery of EWS-ETS fusions and advancing to interventions targeting the cell surface, nucleus, and angiogenesis pathways. Chemotherapy regimens have been developed through collaborative group research employing reliable medications (etoposide, cyclophosphamide, doxorubicin, vincristine, and ifosfamide), even though North America and Europe have different specific protocols.

New treatment options have been made possible by the discovery of EWS-ETS gene fusions and their connection to Ewing’s sarcomas. These include pathways linked to IGF1 and mTOR, as well as suppression of the fusion gene or its protein product.

Among promising innovations are tyrosine kinase inhibitors, currently under investigation as combination and maintenance therapies. Other treatment approaches alter DNA damage, cell cycle progression, and apoptotic pathways while focusing on the EWSR1-FLI1 fusion oncoprotein. Additionally, immunotherapeutic strategies, including CAR-T-cell treatment that targets GD2, have a lot of promise [60,61].

## 8. Future Perspectives

In recent years, changes in treatment protocols introduced in paediatric oncology and the development of new radiological techniques have led to a significant improvement in treatment outcomes.

The field of paediatric cancer treatment is rapidly evolving due to innovative approaches. Some of the prospects include the development of targeted therapies, immunotherapy, genetic therapy, nanotechnology, combined treatments, clinical studies, and multidisciplinary teams.

These developments aim to improve the results and quality of life for young patients affected by osteosarcoma and Ewing sarcoma. These advancements will help identify specific genetic mutations in tumours, improve treatment efficiency, and reduce the risk of recurrence (Figure 9).

The precision and effectiveness of bone tumour diagnosis are being improved by developments in deep learning algorithms and cell segmentation techniques. Osteosarcoma cell pictures can be segmented to provide a reference base for clinical diagnosis and treatment by better understanding the cells’ morphological characteristics, structure, and possible growth and metastasis pathways. Convolutional neural networks and conventional techniques are the two categories of current medical picture segmentation processing techniques [62,63]. An important development in medical imaging technology, especially in the detection of osteosarcoma, is the creation of some technologies, such as the “Twin Swin Transformer” with the multi-scale feature fusion approach. This technique improved osteosarcoma cell segmentation accuracy and efficiency by utilising high-resolution imaging data and sophisticated computational approaches, which will ultimately improve patient care and clinical decision-making [64,65].

The ever-increasing evolution of technology has meant that in the diagnosis of bone tumours and in the diagnosis of osteosarcoma, in particular, artificial intelligence software is coming to our aid, which can help doctors in the diagnosis and localisation of neoplasia. Below is a graph illustrating the positive correlation between the technological evolution of radiodiagnostic imaging over time and the development of the treatments applied to the two pathologies described (Figure 10).

More information will be added to the system as image-processing tools advance, enabling us to create a multi-scale segmentation technique. This will increase segmentation accuracy and help us better resolve segmentation mistakes brought on by minute greyscale changes between the surrounding tissue and the tumour tissue.

In clinical research, more work is still required before AI imaging investigations in musculoskeletal oncology can demonstrate enough evidence to be applied in routine treatment.

However, some AI technologies will undoubtedly be used in practice in the future if strict methodological guidelines are followed. Automating time-consuming, low-value tasks will free up medical time and most likely enhance radiologists’ performance [66,67,68].

In conclusion, prompt diagnosis and management by experienced personnel can make a difference in the outcome of these young patients.

## 9. Conclusions

In conclusion, paediatric bone neoplasms represent a distinct and complex subset of sarcomas, necessitating a nuanced understanding of their biological behaviour and pathophysiology.

Although these tumours are relatively rare, they present considerable challenges in terms of diagnosis, treatment, and long-term management. Collaboration among pathologists, oncologists, radiologists, and surgical teams is essential for the development of personalised treatment plans that are both effective and minimally invasive. This collaborative model ensures that all aspects of care, from initial diagnosis to treatment and follow-up, are meticulously coordinated, thus optimising patient outcomes and enhancing the quality of life for those affected by these challenging malignancies.

Ultimately, continued research and clinical advancements will be vital in further improving the prognosis for paediatric patients facilitating the development of more targeted therapies and more precise, individualised treatment strategies.

## Figures and Tables

**Figure 1 cancers-17-00451-f001:**
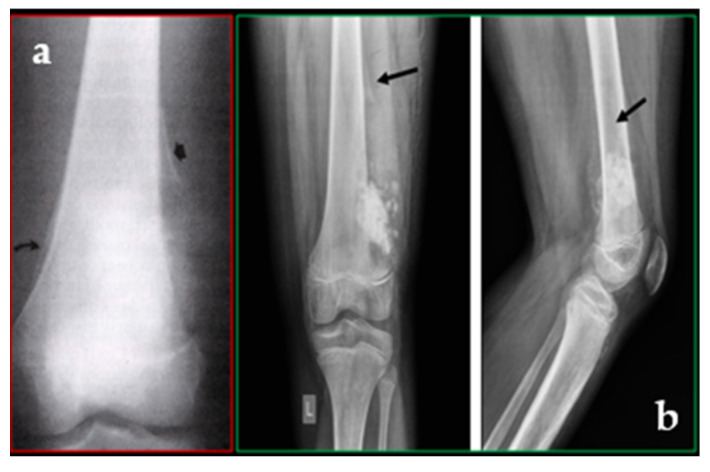
(**a**) Osteosarcoma. Radiographic examination shows eccentric erosive alteration interesting the medial contour of the femoral metaphysis, with presence of ill-defined tumefaction defined extrinsic to the adjacent soft parts. In the context of the bony lesion, a dual component is osteosclerotic and osteolytic (mixed form). It is associated with longitudinal periosteal reaction on the inner side (arrow curve) and with the classic appearance of triangular salience (“the Codman’s triangle”) on the outer slope (straight arrow). (**b**) A 12-year-old boy has osteosarcoma at the distal left femur, characterised by a broad zone of transition, osteoid matrix, and an active periosteal reaction (Codman triangle-arrows). Radiopaedia.org.

**Figure 2 cancers-17-00451-f002:**
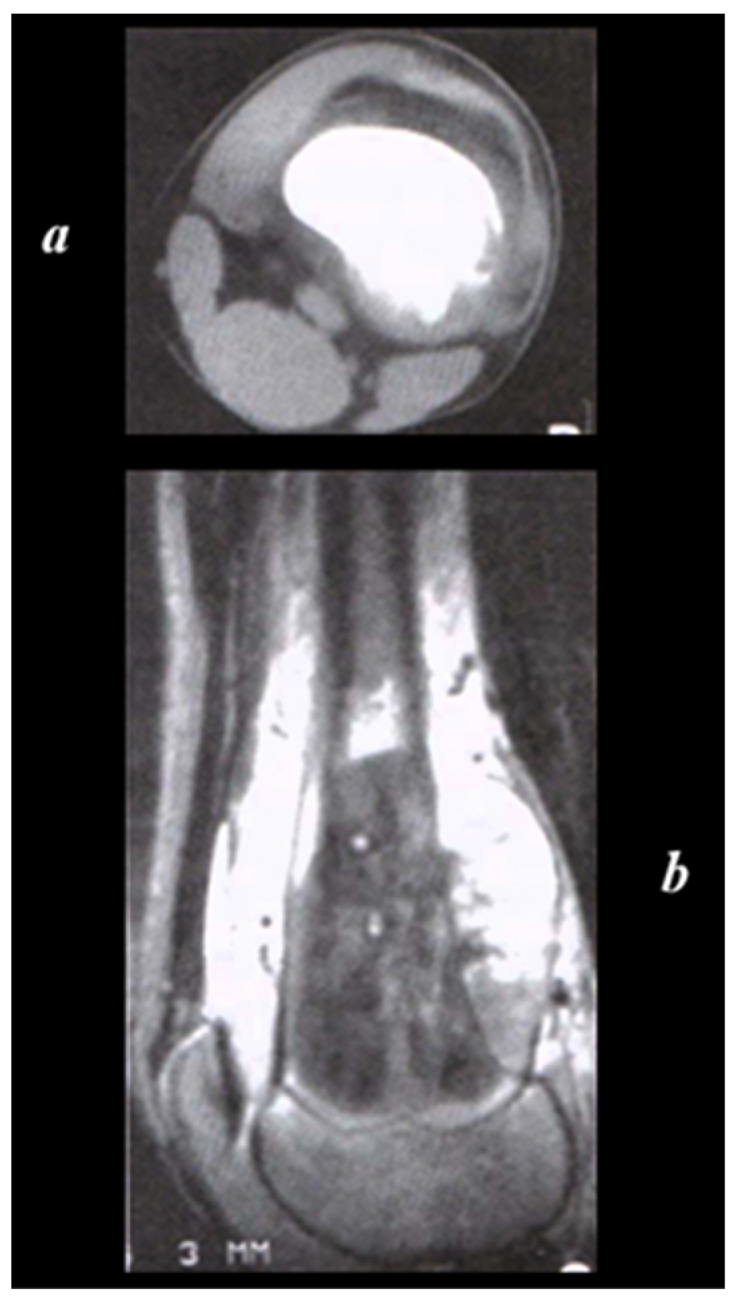
The structural features of the lesion and its encroachment into the paraosteal soft parts are well documented by CT scan (**a**). MRI (**b**) in sagittal section in T2-weighted sequence, showing the occupation of the medullary canal (with clear delimitation in correspondence of the growth cartilage) by the lesion, which emits a low-intensity signal where the osteosclerotic component prevails and high-intensity signal where the osteolytic component prevails. The ‘sleeve-like’ encroachment into the soft parts is evident. Radiopaedia.org.

**Figure 3 cancers-17-00451-f003:**
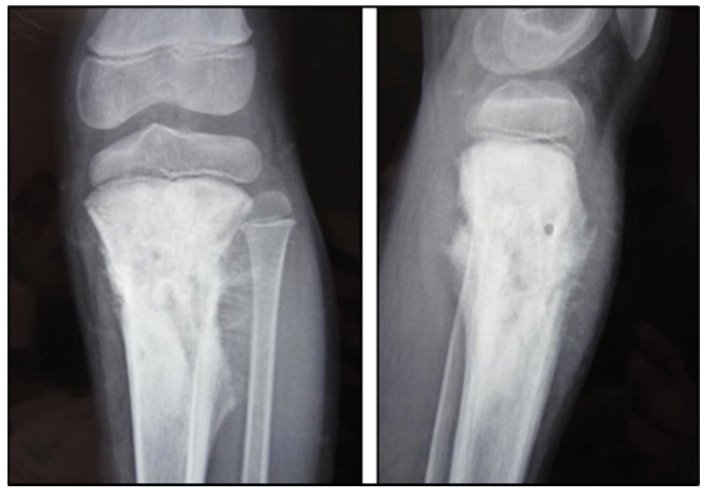
A 9-year-old child’s left knee joint and leg were radiographed in AP and lateral projections. A sclerotic lesion involving the diametaphyseal region of the tibia with a broad zone of transition, osteoid matrix, periosteal elevation (Codman’s Triangle), and the distinctive “Sunburst” type of periosteal reaction is seen on radiographs. There is no discernible intra-articular extension or soft tissue involvement. Radiopaedia.org.

**Figure 4 cancers-17-00451-f004:**
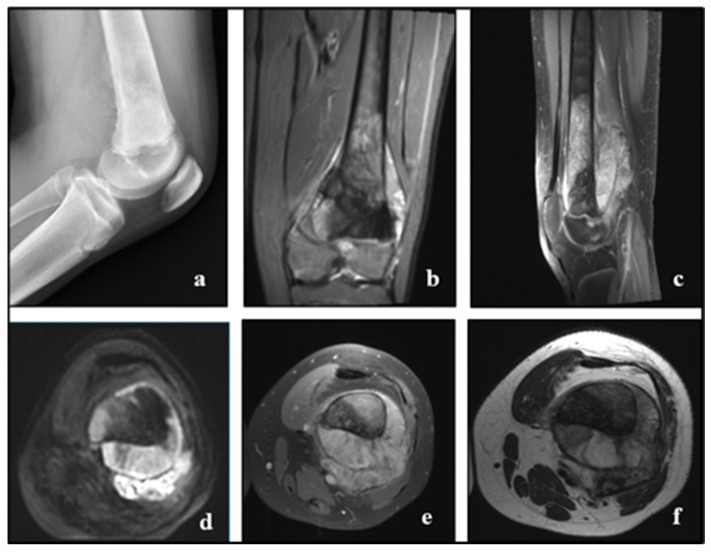
OS in a male of 15 years old: the radiography shows a bone lesion at the distal femoral metaphysis that does not cross the physis, extends in various directions, and causes destruction and loss of distinction between bone marrow and cortical bone, aggressive periosteal reaction of sunburst type and Codman’s triangle, and invasion of adjacent soft tissue (**a**). MRI (coronal STIR (**b**), sagittal T1 + C (**c**), axial DWI (**d**), axial STIR (**e**), and T2-W (**f**)) helps visualise the characteristics previously noted on the radiograph, showing tumor infiltration in the medullary canal and the metaphysis of the distal femur, not crossing the growth plate, with aggressive periosteal reaction (Codman’s triangle), invasion, and formation of adjacent soft tissue mass, restricted diffusion, and strong contrast enhancement after injection. Radiopaedia.org.

**Figure 5 cancers-17-00451-f005:**
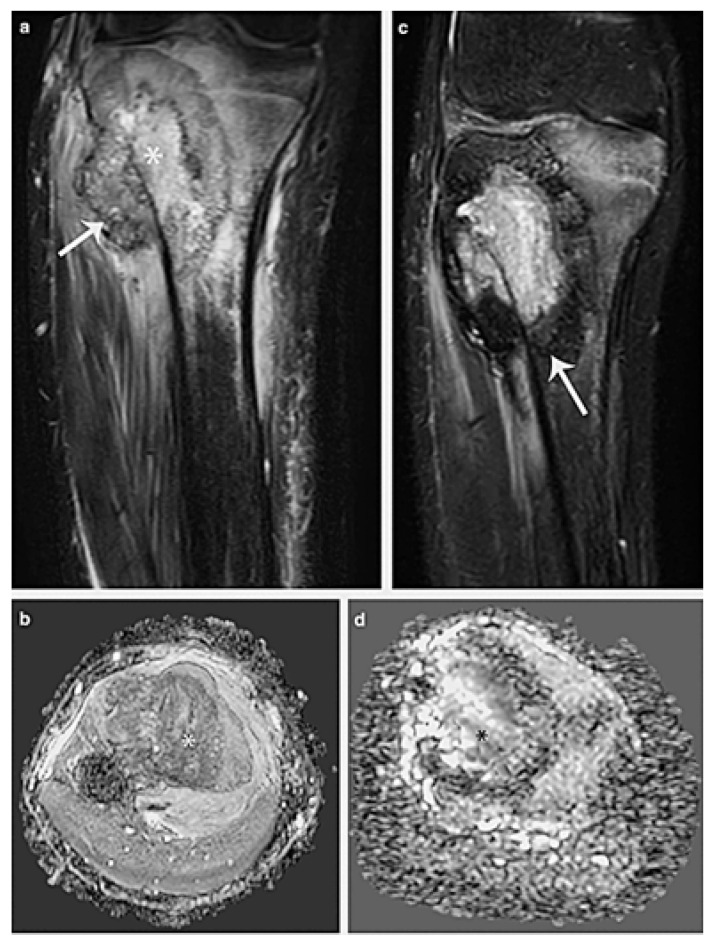
A 13-year-old boy has osteosarcoma affecting his proximal metaphysis of the tibia. (**a**) Soft-tissue mass (arrow) and a heterogeneous bone lesion (*) are seen on a coronal fat-suppressed T2-weighted MR picture. (**b**) The tumour’s diffusion restriction (*) is visible in the axial ADC MR picture. (**c**,**d**) MRI after chemotherapy. (**c**) A peripheral hypointense rim in the tumour that suggests calcification can be seen on the coronal fat-suppressed T2-weighted MR image (arrow). (**d**) When compared to the original research, the axial ADC MR image demonstrates a reduction in the diffusion restriction areas (*), suggesting a positive response to treatment. Narejos Clemente et al. [35].

**Figure 6 cancers-17-00451-f006:**
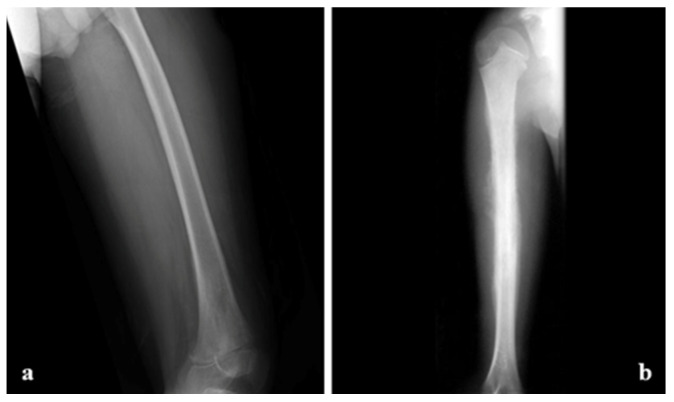
(**a**) There is ill-defined lucency in the distal femur, along with an onion-skin periosteal reaction and a broad zone of transition. (**b**) Humerus X-ray showing cortical erosions, bone loss, and a poor transition to normal bone. In this instance, the centre of the tumour seems to be a sunburst because of perpendicular spiculations. The tumour’s edge, which is not noticeable in this instance, has a laminated appearance that resembles onion skin. Bone sclerosis is rare. There is no distinct soft tissue mass in this instance. Radiopaedia.org.

**Figure 7 cancers-17-00451-f007:**
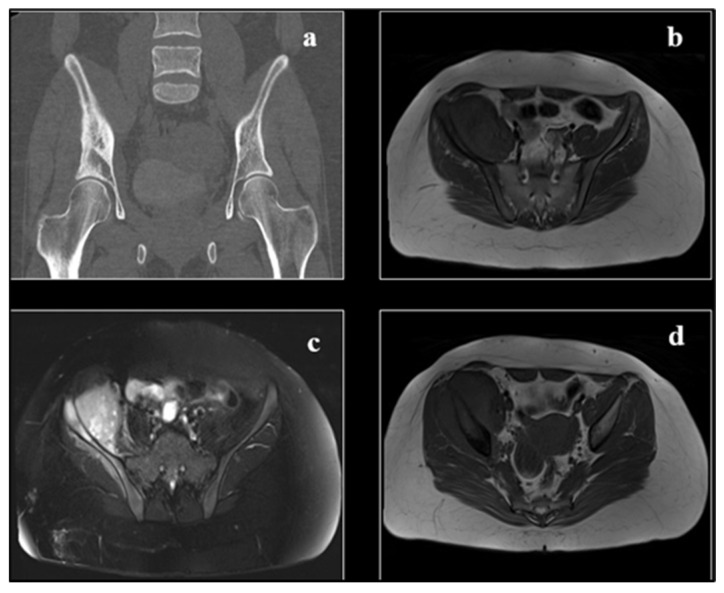
(**a**) CT image demonstrating cortical destruction, medullary extension, and periosteal reaction of an osteolytic aggressive lesion involving the right iliac bone. Large soft tissue mass that is connected to the right iliacus muscle. (**c**) T2 bright heterogeneous signal with heterogeneous post-contrast enhancement (**d**); (**b**) T1 iso to low signal. Along with the infiltration of the iliacus muscle and, to a lesser extent, the gluteus minimus muscle, the associated soft tissue component is observed penetrating and degrading the iliac bone. Radiopaedia.org.

**Figure 8 cancers-17-00451-f008:**
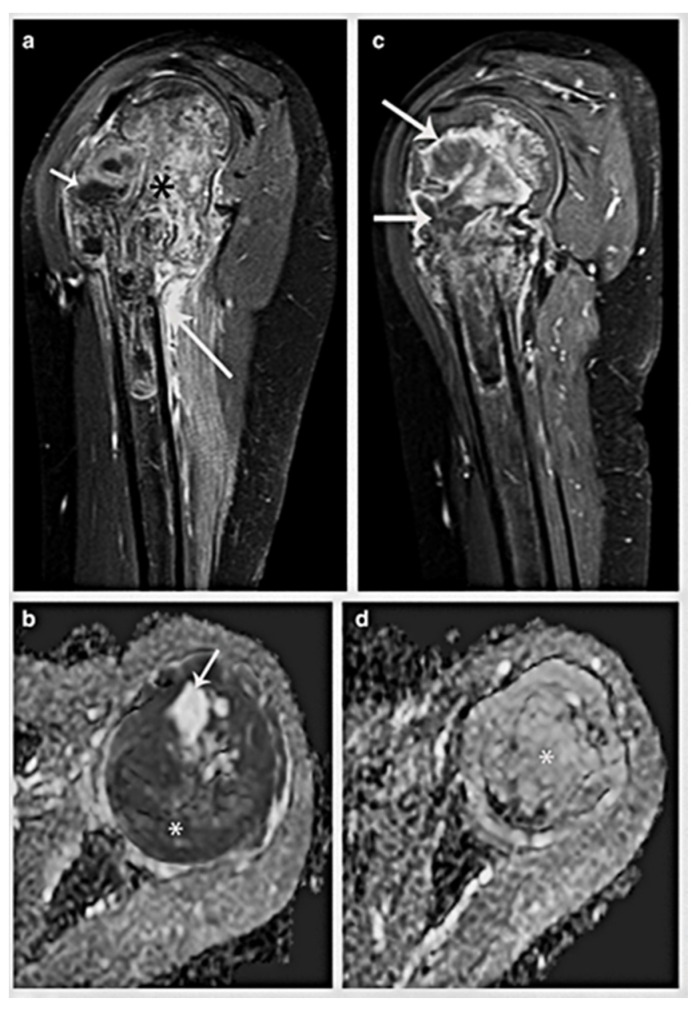
A 12-year-old child has Ewing sarcoma (*) in the proximal left humerus. (**a**) Heterogeneous mass (*) concentrated in the proximal humeral metaphysis that extends to the epiphysis and proximal diaphysis and contains foci of internal necrosis (short arrow) may be seen on a sagittal contrast-enhanced fat-suppressed T1-weighted MR imaging. Take note of the humerus’s significant varus angulation from the prior fracture (long arrow). (**b**) The axial ADC MR image displays no limitation of the necrotic foci (arrow) but limited spread of the cellular regions. (**c**) Following treatment, a sagittal contrast-enhanced fat-suppressed T1-weighted MR image demonstrates a decrease in mass volume as necrotic regions (arrows) rise. (**d**) Following treatment, the axial ADC MR picture reveals no limited diffusion (*). The conventional MRI sequences indicate good treatment response. Narejos Clemente et al. [35].

**Figure 9 cancers-17-00451-f009:**
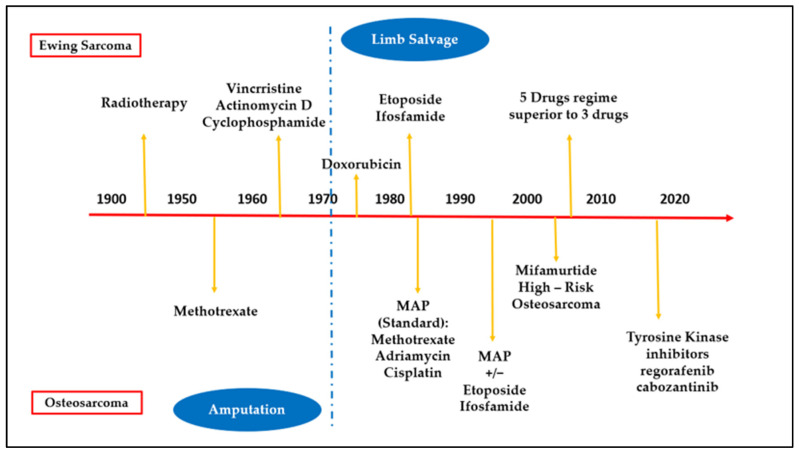
An illustration showing how osteosarcoma and Ewing’s sarcoma have been treated historically until recent times.

**Figure 10 cancers-17-00451-f010:**
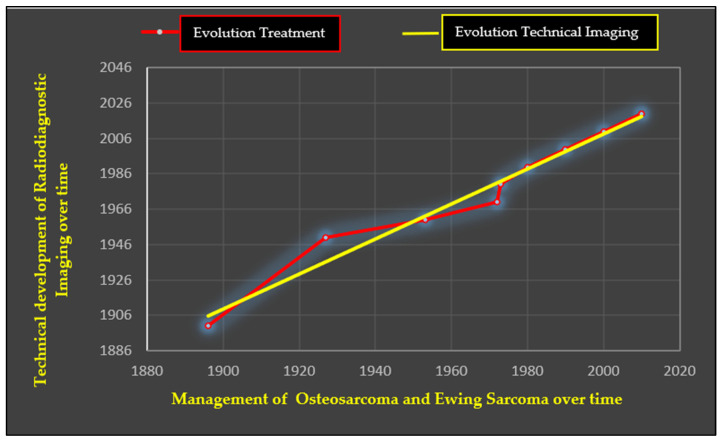
The positive correlation between evolution of radiodiagnostic imaging technology (red line), starting with the radiography of J. Hall Edwards, 1896 to the present, with a future perspective and evolution of treatments (yellow line) used to treat malignant neoplasms under investigation since the late 1880s/1900s.

**Table 1 cancers-17-00451-t001:** Genetic syndromes commonly associated with osteosarcoma and Ewing’s sarcoma.

Osteosarcoma	Ewing Sarcoma
Li–Fraumeni Syndrome	Li–Fraumeni Syndrome
Rothmund–Thomson Syndrome and other RECQL homolog-associated syndromes	Neurofibromatosis
Retinoblastoma	Rothmund–Thomson Syndrome
Hereditary multiple exostoses	//

**Table 2 cancers-17-00451-t002:** Different characteristics of neoplasms, osteosarcoma, and Ewing sarcoma. *International Journal of Molecular Sciences, May 2023* [17].

	Osteosarcoma	Ewing Sarcoma
Age at onset	10–14 years	<20 years (80%)
Sex (M:F)	1.6:1	1.5:1
Predisposing factors and associated syndromes	Retinoblastoma, Paget’s disease, Li–Fraumeni syndrome, Rothmund–Thomson syndrome, and Bloom’s syndrome	None
Location	80% extremities, 20% axial skeleton	Large bones are involved in 50–60% and the axial skeleton in 45%, with greater soft tissue involvement
Bone involvement	Long bone metaphysis, such as that of the femur, tibia, and humerus	Diaphysis of large bones Flat bones
Clinical manifestations	Swelling and masses and later pain	Pain, swelling, masses, fever in 20% and systemic symptoms
Metastasis localisation	Lung and bones	Lung, bones, and bone marrow
Therapy	Phase I: Preoperative chemotherapy Phase II: Surgery Phase III: Postoperative chemotherapy	Phase I: Preoperative chemotherapy Phase II: Surgery and/or radiotherapy Phase III: Postoperative chemotherapy and/or radiation therapy and/or autologous hematopoietic progenitor cell transplantation
Outcomes predictors	Tumour size, axial skeleton involvement, metastasis at the time of diagnosis, increased alkaline phosphatase levels, and a poor response to preoperative chemotherapy with less than 90% necrosis	Metastasis at diagnosis, more important, axial location, tumour volume greater than 200 mL, maximal diameter exceeding 8 cm, older age, male sex, elevated LDH, gene expression profile with p53, Ki67 overexpression, and 16q loss, and poor response to preoperative chemotherapy with less than 100% necrosis
Survival rate	Undisseminated ranges from 60 to 74%, while disseminated are around 30%	Undisseminated ranges from 60 to 75%, while disseminated ranges from 20 to 30%

**Table 3 cancers-17-00451-t003:** Imaging modalities and considerations. “*Radiographic*” Volume 42, Issue 4 July–August 2022 [19].

Modality	Clinical Indications	Advantages	Disadvantages
Radiography	First-line modality for screening patients suspected of having bone disease Screening for local recurrence and complications related to surgical reconstruction	Readily available Does not require sedation	Limited soft-tissue detail Unable to define medullary extent Radiation exposure
Tumour MRI	Preferred for tumour characterisation and preoperative planning (biopsy and en bloc resection)	Definition of intramedullary involvement and extent characterisation of extraosseous bulk and surrounding tissue involvement	May not be readily available May require patient sedation
Whole-body MRI	Identification of potential sites of disease, often coupled with PET	Detection of additional sites of disease	Inadequate evaluation of pulmonary disease May not be readily available May require patient sedation
Tumour CT	Not routinely performed in children unless tumour is located at anatomically complex sites or patient requires custom reconstruction	Delineation of cortical destruction Characterisation of intratumour calcification and ossification	Limited soft-tissue detail Unable to define medullary extent Radiation exposure
Chest CT	Identification of pulmonary Metastases	Discernment of need for metastasectomy	Radiation exposure
Whole-body ^Com^Tc MDP bone scintigraphy	Identification of sites of increased blood flow and new bone formation * (60)	Detection of additional disease sites	Lacks anatomic detail; thus, additional imaging such as radiography, SPECT, or MRI is required to further distinguish between tumour and false-positive findings (e.g., injury-related healing) Radiation exposure
Whole-body FDG PET	Identification of sites of increased glucose metabolism *	Detection of additional disease sites SUV alterations can predict chemotherapy response; Outperforms bone scintigraphy in detection of osseous metastases	Lacks anatomic detail; thus, additional imaging such as CT and MRI are required Lacks consensus SUV cutoff values between good and poor responders, which differ depending on imaging protocols and postprocessing sampling methods Radiation exposure

Note. At the authors’ institution (Children’s Hospital of Philadelphia), MRI examinations of the primary tumour are routinely performed with intravenous contrast material, and CT examinations of the chest are routinely performed without intravenous contrast material. Numbers in parentheses are reference numbers. FDG = fluorine 18 (F) fluorodeoxyglucose, MDP = methylene diphosphonate, SUV = standardised uptake value. * Increased radiotracer update is nonspecific and may or may not be related to osteosarcoma.

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
