# Peer review of "Analysis and Historical Evolution of Paediatric Bone Tumours: The Importance of Early Diagnosis in the Detection of Childhood Skeletal Malignancies"

_cancers, 2025, doi:10.3390/cancers17030451_

Round 1
Reviewer 1 Report
Comments and Suggestions for Authors
In the present work Iacobellis et al. reviewed the literature concerning the current knowledge on pediatric musculoskeletal primary tumors. The present review is interesting from its perception. Although the idea is not novel, since there are numerous reviews concerning the clinical aspects of primary bone tumors in children, still such works are always useful for these types of tumors.
One thing that should be clarified is the “early” diagnosis. It gets a bit confusing since the authors present the clinical tools for diagnosis, mainly imaging, but all this is at diagnosis. The authors should clarify how their report concerns the “early” diagnosis of musculoskeletal tumors. Since, the talk is about “early”, some molecular or genomic/transcriptomic biomarkers could be also discussed. Further on, the authors should explain the meaning of early diagnosis, since the main clinical findings are at the time of clinical presentation, where the disease, sometimes, is already settled.
In addition, there is another musculoskeletal primary pediatric tumor of great interest, which is rhabdomyosarcoma. The authors should also include this type of tumor in their review, in order to have a complete picture of primary musculoskeletal tumors. The title suggests a review on pediatric bone tumors, but Ewing sarcomas could be both a bone as well as soft tissue malignancies. Therefore, the authors could modify their title to “musculoskeletal” tumors. Same here, concerning rhabdomyosarcoma, it would be useful to present the clinical characteristics as well as some genomic/transcriptomic characteristics with relation to early diagnosis.
Overall, this is an interesting review, yet it would be useful to amend it with some more information in order to be more complete.
Author Response
Dear Reviewer1,
Thanks for the comments.
1) Comment: A new section on the early diagnosis of bone tumors evaluated in the review has been added, detailing clinical and genomic aspects and specifying some possible biomarkers useful for early diagnosis and prognosis: “3. Early diagnosis of Osteosarcoma and Sarcoma di Ewing” and “Table 1. Genetic syndromes commonly associated with Osteosarcoma and Ewing's Sarcoma”.
2) Comment: Although rhabdomyosarcoma is one of the most frequent malignant tumours in the paediatric population, as it is a soft tissue tumour, it is outside the scope of this review, which focuses on the most frequent malignant bone tumours in children. His advice, much appreciated however, paved the way for future work that will only look at paediatric muscle tumours.
Best regards.
Reviewer 2 Report
Comments and Suggestions for Authors
Dear Authors,
Enough good elaborated manscrript about this relatively limited topic in the oncology field.
However, I have some objections for the manuscript:
1) Title. Please, add some words regarding the history, because you are using 9 "old" references. Doesnt matter that they are probably good, they do not fit to nowadays manuscripts. But, - You can add some words to the Title, for instance: "...skeletal malignancies with some historical aspects" or probabaly better. Then you can keep these References which otherwise should be removed or changed by the others;
2) Before Introduction please include some common paragraph with methodological aspects and a plan of the review. Actually, move please the first part of your Conclusions plus give the aim of this review, data bases searched, inclusion/excludion criteria for the references, time when the work was done (from...to)
3) 4.1 and 5.1 titles "pathology" is too large, make these places more precise and decipher the described content of the subsection in the subtitle. Othwerwise, sorry, about the Pathology one can write numerous books and never end... These subtitles are not valid simply. Add, for instance, ...etiology, pathogenesis and so on..."
4) check everywhere your small Grammar mistakes and develop uniform style for the references cited in the text and for Figs. the same... Sorry, its very irritating to read an interesting mansucript with carelessness mistakes...
5) indicate for the Fig. 9 is this developed by the authors or from what other source is it adopted.
6) Conclusions. Remove plan of the manuscript from this place and shorten, and make the conclusions more precise.
7) References - see the comment above.
Author Response
Dear Reviewer2,
Thanks for the comments.
1) Comment: The title has been revised with historical notes, being a review, an evaluation of even dated clinical studies was carried out to assess the evolution of imaging with respect to early diagnosis and treatment, over time.
Title: Analysis and historical evolution of paediatric bone tumours: the importance of early diagnosis in the detection of childhood skeletal malignancies.
2) Comment: Following the review standards of the journal Cancers, there is no paragraph in which to include research methods. However, we follow your careful advice and have inserted, after the introduction because it is not possible to insert it before, a ‘Materials and Methods’ paragraph in which we explain the method of research and drafting the review.
3) Comment: Sections 4.1 and 5.1, which became 6.1 and 7.1 after the extension of the revision, have been replaced with “Pathological subtypes” for both bone tumours.
4) Comment: a revision of grammatical errors in the text was carried out and the references were sorted following a sequence of subject order and numerical order. Figure references (Figures) and table references (Tables) have also been revised as required.
5) Comment: Figure 9 is inspired by other images that determine the evolution of imaging and treatment of pathology over time but is original and completely revised by the team that developed this review.
6) Comment: Conclusions have been shortened according to the suggestions of the reviewer and the aims of the reviewer have been better clarified.
7) Revised as requested.
Best regards
Round 2
Reviewer 1 Report
Comments and Suggestions for Authors
The authors have addressed my previous comments. Their work has merit for publication in its present form with one caveat that needs to be addressed. Their similarity index is 21%, which I think is high. Although, sometimes in such works terminology adds up to the similarity index, I think the obtained percentage with filtering the terms out (<1%). Thus, I would suggest to the authors to rephrase parts of their work in order to obtain a lower similarity index.
Author Response
Dear Reviewer,
Thank you for highlighting that the similarity index is high, unfortunately in the reviews it happens unintentionally.
As indicated, we have reworded these parts and highlighted them in green.
Thank you and kind regards